# Safety and Effectiveness of Chemotherapy in Elderly Biliary Tract Cancer Patients

Takeshi Okamoto [1,*], Tsuyoshi Takeda [1], Takashi Sasaki [1], Tsuyoshi Hamada [1,2], Takafumi Mie [1], Takahiro Ishitsuka [1], Manabu Yamada [1], Hiroki Nakagawa [1], Tatsuki Hirai [1], Takaaki Furukawa [1], Akiyoshi Kasuga [1], Masato Ozaka [1] and Naoki Sasahira [1]

1   Department of Hepato-Biliary-Pancreatic Medicine, Cancer Institute Hospital of Japanese Foundation for Cancer Research, 3-8-31 Ariake, Koto-ku, Tokyo 135-8550, Japan; tsuyoshi.takeda@jfcr.or.jp (T.T.); takashi.sasaki@jfcr.or.jp (T.S.); tys.hamada@gmail.com (T.H.); takafumi.mie@jfcr.or.jp (T.M.); takahiro.ishitsuka@jfcr.or.jp (T.I.); manabu.yamada@jfcr.or.jp (M.Y.); hiroki.nakagawa@jfcr.or.jp (H.N.); tatsuki.hirai@jfcr.or.jp (T.H.); takaaki.furukawa@jfcr.or.jp (T.F.); akiyoshi.kasuga@jfcr.or.jp (A.K.); masato.ozaka@jfcr.or.jp (M.O.); naoki.sasahira@jfcr.or.jp (N.S.)

2   Department of Gastroenterology, Graduate School of Medicine, The University of Tokyo, 7-3-1 Hongo, Bunkyo-ku, Tokyo 113-8654, Japan

*   Correspondence: takeshi.okamoto@jfcr.or.jp

**Abstract:** The safety and effectiveness of chemotherapy in elderly patients with biliary tract cancer (BTC) remain unclear. Therefore, we retrospectively reviewed patients who underwent chemotherapy for locally advanced, metastatic, or recurrent BTC at our institution from January 2016 to December 2021. Of the 283 included patients, 91 (32.5%) were aged 75 years or older when initiating chemotherapy. Elderly patients were more likely than non-elderly patients to receive monotherapy with gemcitabine or S-1 (58.7% vs. 9.4%, $p < 0.001$) and were less likely to experience grade 3–4 toxicities (55.4% vs. 70.2%, $p = 0.015$). The rates of termination due to intolerance (6.5% vs. 5.8%, $p = 0.800$) and transition to second-line chemotherapy (39.1% vs. 40.3%, $p = 0.849$) were similar between groups. In the overall cohort, age was not an independent predictor of overall survival (OS). Within the elderly cohort, there were no differences in severe adverse events between patients receiving monotherapy and combination therapy (50.0% vs. 63.2%, $p = 0.211$). Median OS was longer in the combination therapy group (10.4 vs. 14.1 months; $p = 0.010$); however, choice of monotherapy was not an independent predictor of overall survival. Monotherapy appears to be a viable alternative in selected elderly BTC patients.

**Keywords:** age; cholangiocarcinoma; monotherapy; combination chemotherapy; tolerability; gemcitabine; cisplatin; S-1

## 1. Introduction

Biliary tract cancer (BTC) is a collective term that refers to a heterogenous group of malignancies arising in the biliary tree, including intrahepatic and extrahepatic (perihilar or distal) cholangiocarcinomas, gallbladder cancer, and sometimes ampullary cancer [1,2]. BTC is primarily a disease of the elderly. As of 2019 in Japan, 89% of patients diagnosed with BTC were aged 65 years or older, 77% were 70 years or older, 64% were 75 years or older, 47% were 80 years or older, and 28% were 85 years or older [3]. There is a similar trend worldwide; for example, the incidence of gallbladder cancer peaks at 85–89 years of age in the United Kingdom [4,5].

While only surgical resection offers a chance for cure, a large majority of BTC cases are unresectable at diagnosis [6]. Chemotherapy must therefore be considered, even in elderly cases. With one notable exception [7], most clinical trials for BTC have avoided imposing an upper age limit to its participants, focusing instead of Eastern Cooperative Oncology Group (ECOG) performance status (PS) [8–17]. Nevertheless, the median age of included patients

is generally about 65 years old, with very little information for patients older than the age of 75 years. We therefore conduct this study to (1) investigate the outcomes of chemotherapy for elderly (aged 75 years and over) and non-elderly BTC patients in the real-world setting and (2) compare the outcomes of monotherapy and combination chemotherapy in elderly BTC patients.

## 2. Materials and Methods

### 2.1. Patients

We conducted a retrospective review of consecutive patients with unresectable (locally advanced, metastatic, or recurrent) BTC who received first-line chemotherapy at our institution between 1 January 2016, and 31 December 2021. For the purposes of this study, BTC included intrahepatic and extrahepatic (perihilar or distal) cholangiocarcinomas, gallbladder cancer, and ampullary cancer. While ampullary cancer is sometimes excluded from clinical trials on BTC due to its unique characteristics, it was included in this study as all chemotherapy regimens for BTC received by the study subjects were also indicated for ampullary cancer in Japan. Data were extracted from a prospectively maintained database. Patients enrolled in clinical trials at any time and distal cholangiocarcinoma patients treated with pancreatic cancer regimens due to initial misdiagnosis as pancreatic cancer were excluded from this study.

### 2.2. Baseline Characteristics

Age, ECOG PS, resectability status, presence and location of metastases, and laboratory data, including tumor markers, were evaluated at the time of diagnosis. Patients aged 75 years and older were considered elderly for the purposes of this study, based on the age distribution of BTC and in accordance with the latest proposal from the Japan Gerontological Society and the Japan Geriatrics Society [18] and with recent reports [19,20]. The modified Glasgow prognostic score (mGPS) was calculated based on serum albumin and C-reactive protein (CRP) at diagnosis, scored as 0 if CRP $\leq 1$ mg/dL, as 1 if albumin $\geq 3.5$ g/dL and CRP $\geq 1$ mg/dL, and as 2 if albumin $\leq 3.5$ g/dL and CRP $\geq 1$ mg/dL [21]. Neutrophil-to-lymphocyte ratio (NLR) was calculated as the ratio of the absolute neutrophil count to the absolute lymphocyte count [22,23].

### 2.3. Chemotherapy

Chemotherapy regimens were selected at the discretion of the oncologist, taking age, general physical condition, cancer status, and other factors into account, and after careful discussions with each patient. Choice of chemotherapy regimen for elderly patients was discussed at department conferences and/or at a multidisciplinary cancer board. Initial dosages were determined based on clinical trials and adjusted for decreased renal function and other relevant factors, but were not reduced solely due to age. Subsequent dosages were reduced based on adverse events, as needed. Adverse events were evaluated based on the National Cancer Institute Common Terminology Criteria for Adverse Events version 4.0 [24]. Chemotherapy was continued until disease progression, patient refusal, intolerable toxicity, conversion surgery, or death.

Contrast-enhanced computed tomography (CT) was performed every 2–3 months, except in cases that developed kidney injury or allergies to contrast media during the follow-up period. Response to chemotherapy was defined as best tumor response on follow-up imaging studies and was evaluated in accordance with the response evaluation criteria in solid tumors (RECIST) guideline (version 1.1) [25]. Overall survival (OS) was defined as the time from the first day of chemotherapy until death from any cause or the last follow-up. Progression-free survival (PFS) was defined as the time from the first day of chemotherapy until death from any cause, disease progression, or the last follow-up. Follow-up data were confirmed up to 31 March 2023.

*2.4. Statistical Analysis*

Categorical variables are shown as absolute numbers and percentages, while continuous variables are shown as medians with ranges. Denominators of ratios were adjusted for missing data. Statistical analyses were conducted using chi-squared or Fisher's exact tests for categorical variables and the Mann–Whitney U test for continuous variables. Kaplan–Meier and log-rank analyses were conducted to evaluate OS and PFS. Cox regression analysis was conducted to investigate factors associated with OS. Multivariate analysis was performed on variables considered significant in univariate analysis, excluding variables that were not known when chemotherapy was started. *p*-values were two-sided and values < 0.05 were considered statistically significant. All statistical analyses were performed using IBM SPSS Statistics ver. 28.0 (IBM Corp., Armonk, NY, USA).

*2.5. Ethical Considerations*

This study was approved by the Institutional Review Board at our hospital (2023-GB-016). Patient consent was waived due to its retrospective design. The study was publicized on the hospital website, allowing patients to opt out of the study without impacting their care.

## 3. Results

A total of 320 patients commenced first-line chemotherapy at our institution during the study period. We excluded 32 cases that participated in clinical trials and 5 cases initially diagnosed as pancreatic cancer and treated with modified FOLFIRINOX. As a result, 283 patients were included in this study.

*3.1. Patient Characteristics*

Baseline characteristics are shown in Table 1. Elderly patients aged 75 years or older were more likely to have worse ECOG PS compared to the non-elderly group (66.3% vs. 81.7% had ECOG PS of 0), and to have distal cholangiocarcinomas (29.3% vs. 12.6%, *p* < 0.001). Other characteristics were similar between groups.

Elderly patients undergoing monotherapy were older (median of 81 vs. 77 years old, *p* < 0.001) and had worse ECOG PS (55.6% vs. 81.6% had ECOG PS of 0) than those who received combination chemotherapy. The maximum age was 89 years old in the monotherapy group and 82 years old in the combination therapy group. No other significant differences in baseline characteristics were observed.

*3.2. Treatment-Related Characteristics*

Despite less patients receiving combination therapy in the elderly group than in the non-elderly group (41.3% vs. 90.6%, *p* < 0.001), no significant differences in responses to first-line chemotherapy were observed (Table 2). A similar number of patients were able to proceed to second-line chemotherapy (39.1% vs. 40.3%, *p* = 0.849), which involved S-1 monotherapy in over 80% of cases in both groups. There was a tendency for non-elderly patients to undergo conversion surgery (3.3% vs. 8.9%, *p* = 0.083), although the difference was not significant.

Within the elderly group, the monotherapy group tended to have a lower overall response rate (2.2% vs. 14.3%, *p* = 0.081) and to have a lower rate of conversion surgery (0% vs. 7.9%, *p* = 0.067); however, the differences were not significant. The monotherapy group was less likely to proceed to second-line therapy (29.6% vs. 52.6%, *p* = 0.026).

*3.3. Adverse Events*

Adverse events are summarized in Table 3. Elderly patients reported less all-grade constipation and nausea/vomiting but were more likely to experience decreased renal function. Non-elderly patients were more likely to experience severe adverse events (grades 3 or 4; 70.2% vs. 55.4%, *p* = 0.015). Specifically, non-elderly patients experienced more severe episodes of leukopenia, neutropenia, and elevated transaminases.

**Table 1.** Baseline characteristics.

| | | | | | | | Elderly Patients | | | |
|---|---|---|---|---|---|---|---|---|---|---|
| | **Non-Elderly** | | **Elderly** | | | **Monotherapy** | | **Combination** | | |
| | (*n* = 191) | | (*n* = 92) | | *p*-Value | (*n* = 54) | | (*n* = 38) | | *p*-Value |
| Age in years, median (range) | 67 | (24–74) | 78.5 | (75–89) | <0.001 | 81 | (75–89) | 77 | (75–82) | <0.001 |
| Male (*n*, %) | 116 | 60.7% | 55 | 59.8% | 0.878 | 35 | 64.8% | 20 | 52.6% | 0.241 |
| Body mass index, median (range) | 20.8 | (13.6–34.4) | 20.6 | (14.1–29.5) | 0.352 | 20.5 | (14.9–28.9) | 21.7 | (14.1–29.5) | 0.168 |
| Performance status, 0/1/2 | 156/34/1 | | 61/29/2 | | 0.013 | 30/22/2 | | 31/7/0 | | 0.027 |
| Primary cancer (*n*, %) | | | | | | | | | | |
|   Intrahepatic | 48 | 25.1% | 15 | 16.3% | 0.095 | 9 | 16.7% | 6 | 15.8% | 0.911 |
|   Extrahepatic (perihilar) | 51 | 26.7% | 28 | 30.4% | 0.512 | 18 | 33.3% | 10 | 26.3% | 0.471 |
|   Extrahepatic (distal) | 24 | 12.6% | 27 | 29.3% | <0.001 | 13 | 24.1% | 14 | 36.8% | 0.185 |
|   Gallbladder | 51 | 26.7% | 18 | 19.6% | 0.190 | 13 | 24.1% | 5 | 13.2% | 0.194 |
|   Ampulla | 17 | 8.9% | 4 | 4.3% | 0.171 | 1 | 1.9% | 3 | 7.9% | 0.303 |
| Cancer status (*n*, %) | | | | | | | | | | |
|   Locally advanced | 33 | 17.3% | 13 | 14.1% | 0.501 | 5 | 9.3% | 8 | 21.1% | 0.110 |
|   Metastatic | 96 | 50.3% | 42 | 45.7% | 0.467 | 26 | 48.1% | 16 | 42.1% | 0.567 |
|   Recurrent | 62 | 32.5% | 37 | 40.2% | 0.200 | 23 | 42.6% | 14 | 36.8% | 0.580 |
| Location of metastases [1] | | | | | | | | | | |
|   Liver | 68 | 35.6% | 32 | 34.8% | 0.893 | 21 | 38.9% | 11 | 28.9% | 0.324 |
|   Lung | 25 | 13.1% | 13 | 14.1% | 0.810 | 5 | 9.3% | 8 | 21.1% | 0.110 |
|   Lymph nodes | 62 | 32.5% | 25 | 27.2% | 0.367 | 15 | 27.8% | 10 | 26.3% | 0.877 |
|   Peritoneal dissemination | 52 | 27.2% | 23 | 25.0% | 0.691 | 14 | 25.9% | 9 | 23.7% | 0.807 |
|   Bone | 6 | 3.1% | 1 | 1.1% | 0.297 | 1 | 1.9% | 0 | 0.0% | >0.999 |
| Laboratory data | | | | | | | | | | |
|   Modified Glasgow prognostic score, 0/1/2 | 127/32/32 | | 52/17/23 | | 0.207 | 27/11/16 | | 25/6/7 | | 0.308 |
|   Neutrophil-to-lymphocyte ratio, median (range) | 2.5 | (0.4–30.7) | 2.7 | (0.7–14.0) | 0.955 | 2.8 | (0.7–14.0) | 2.6 | (1.6–12.3) | 0.943 |
|   CEA, ng/mL, median (range) | 3.3 | (0.5–1398) | 4.0 | (1.1–386) | 0.238 | 4.0 | (1.1–386) | 4.0 | (1.2–358) | 0.643 |
|   CA19-9, U/mL, median (range) | 159 | (2–50,000) | 177 | (2–50,000) | 0.739 | 159 | (2–50,000) | 177 | (2–50,000) | 0.883 |

CA19-9: carbohydrate antigen 19-9; CEA: carcinoembryonic antigen. [1] Some patients had metastases to multiple locations, while others had none.

**Table 2.** Treatment-related characteristics.

| | | | | | | | Elderly Patients | | | |
|---|---|---|---|---|---|---|---|---|---|---|
| (*n*, %) | **Non-Elderly** | | **Elderly** | | | **Monotherapy** | | **Combination** | | |
| | (*n* = 191) | | (*n* = 92) | | *p*-Value | (*n* = 54) | | (*n* = 38) | | *p*-Value |
| First-line chemotherapy | 191 | 100.0% | 92 | 100.0% | - | 54 | 100.0% | 38 | 100.0% | |
|   Combination therapy | 173 | 90.6% | 38 | 41.3% | <0.001 | | | | | |
|     Gemcitabine + cisplatin + S-1 | 19 | 9.9% | 1 | 1.1% | 0.006 | | | 1 | 2.6% | |
|     Gemcitabine + cisplatin | 150 | 78.5% | 34 | 37.0% | <0.001 | | | 34 | 89.5% | |
|     Gemcitabine + S-1 | 4 | 2.1% | 3 | 3.3% | 0.686 | | | 3 | 7.9% | |
|   Monotherapy | 18 | 9.4% | 54 | 58.7% | <0.001 | | | | | |
|     Gemcitabine | 17 | 8.9% | 36 | 39.1% | <0.001 | 36 | 66.7% | | | |
|     S-1 | 1 | 0.5% | 18 | 19.6% | <0.001 | 18 | 33.3% | | | |
| Response to first-line chemotherapy | | | | | | | | | | |
|   Complete response | 1 | 0.5% | 0 | 0.0% | >0.999 | 0 | 0.0% | 0 | 0.0% | - |
|   Partial response | 22 | 11.5% | 6 | 6.5% | 0.187 | 1 | 1.9% | 5 | 13.2% | 0.078 |
|   Stable disease | 112 | 58.6% | 48 | 52.2% | 0.304 | 26 | 48.1% | 22 | 57.9% | 0.357 |
|   Progressive disease | 40 | 20.9% | 26 | 28.3% | 0.173 | 18 | 33.3% | 8 | 21.1% | 0.198 |
|   Not evaluated | 16 | 8.4% | 12 | 13.0% | 0.218 | 9 | 16.7% | 3 | 7.9% | 0.347 |
|   Overall response rate | | 13.1% | | 7.5% | 0.188 | | 2.2% | | 14.3% | 0.081 |
|   Disease control rate | | 77.1% | | 67.5% | 0.103 | | 60.0% | | 77.1% | 0.104 |
| Reason for termination of first-line chemotherapy | | | | | | | | | | |
|   Disease progression | 144 | 75.4% | 77 | 83.7% | 0.114 | 45 | 83.3% | 32 | 84.2% | 0.911 |
|   Intolerance | 11 | 5.8% | 6 | 6.5% | 0.800 | 5 | 9.3% | 1 | 2.6% | 0.395 |
|   Conversion surgery | 17 | 8.9% | 3 | 3.3% | 0.083 | 0 | 0.0% | 3 | 7.9% | 0.067 |
|   Patient refusal | 7 | 3.7% | 1 | 1.1% | 0.220 | 1 | 1.9% | 0 | 0.0% | >0.999 |
|   Treatment ongoing | 2 | 1.0% | 1 | 1.1% | >0.999 | 0 | 0.0% | 1 | 2.6% | 0.413 |
|   Other | 10 | 5.2% | 4 | 4.3% | | 3 | 5.6% | 1 | 2.6% | |
| Second-line chemotherapy | 77 | 40.3% | 36 | 39.1% | 0.849 | 16 | 29.6% | 20 | 52.6% | 0.026 |
|   Gemcitabine + cisplatin | 5 | 2.6% | 2 | 2.2% | | 0 | 0.0% | 2 | 5.3% | 0.168 |
|   Gemcitabine + S-1 | 4 | 2.1% | 1 | 1.1% | | 0 | 0.0% | 1 | 2.6% | 0.413 |
|   Gemcitabine | 0 | 0.0% | 2 | 2.2% | | 2 | 3.7% | 0 | 0.0% | 0.510 |
|   S-1 | 64 | 33.5% | 31 | 33.7% | | 14 | 25.9% | 17 | 44.7% | 0.060 |
|   Other | 4 | 2.1% | 0 | 0.0% | | 0 | 0.0% | 0 | 0.0% | - |

**Table 3.** Adverse events—overall cohort.

| (*n*, %) | All Grades | | | | | | Grades 3–4 | | | | | |
|---|---|---|---|---|---|---|---|---|---|---|---|---|
| | Non-Elderly | | Elderly | | | | Non-Elderly | | Elderly | | | |
| | (*n* = 191) | | (*n* = 92) | | *p*-Value | | (*n* = 191) | | (*n* = 92) | | *p*-Value | |
| All adverse events | 191 | 100.0% | 92 | 100.0% | - | | 134 | 70.2% | 51 | 55.4% | 0.015 | |
| Hematologic adverse events | | | | | | | | | | | | |
| Leukopenia | 141 | 73.8% | 60 | 65.2% | 0.135 | | 55 | 28.8% | 12 | 13.0% | 0.004 | |
| Neutropenia | 152 | 79.6% | 67 | 72.8% | 0.203 | | 103 | 53.9% | 33 | 35.9% | 0.004 | |
| Anemia | 186 | 97.4% | 90 | 97.8% | >0.999 | | 42 | 22.0% | 19 | 20.7% | 0.798 | |
| Thrombocytopenia | 148 | 77.5% | 70 | 76.1% | 0.793 | | 14 | 7.3% | 6 | 6.5% | 0.804 | |
| Febrile neutropenia | 0 | 0.0% | 0 | 0.0% | - | | 0 | 0.0% | 0 | 0.0% | - | |
| Non-hematologic adverse events | | | | | | | | | | | | |
| Stomatitis | 45 | 23.6% | 9 | 9.8% | 0.006 | | 1 | 0.5% | 0 | 0.0% | >0.999 | |
| Decreased appetite | 28 | 14.7% | 21 | 22.8% | 0.089 | | 1 | 0.5% | 1 | 1.1% | 0.545 | |
| Diarrhea | 32 | 16.8% | 16 | 17.4% | 0.894 | | 0 | 0.0% | 0 | 0.0% | - | |
| Constipation | 153 | 80.1% | 54 | 58.7% | <0.001 | | 0 | 0.0% | 0 | 0.0% | - | |
| Nausea/vomiting | 107 | 56.0% | 30 | 32.6% | <0.001 | | 1 | 0.5% | 0 | 0.0% | >0.999 | |
| Peripheral neuropathy | 64 | 33.5% | 14 | 15.2% | 0.001 | | 0 | 0.0% | 0 | 0.0% | - | |
| Alopecia | 9 | 4.7% | 5 | 5.4% | 0.776 | | 0 | 0.0% | 0 | 0.0% | - | |
| Fatigue | 166 | 86.9% | 70 | 76.1% | 0.022 | | 1 | 0.5% | 2 | 2.2% | 0.248 | |
| Elevated transaminases | 173 | 90.6% | 82 | 89.1% | 0.703 | | 25 | 13.1% | 5 | 5.4% | 0.050 | |
| Decreased renal function | 35 | 18.3% | 28 | 30.4% | 0.022 | | 0 | 0.0% | 1 | 1.1% | 0.325 | |
| Interstitial pneumonitis | 0 | 0.0% | 2 | 2.2% | 0.105 | | 0 | 0.0% | 2 | 2.2% | 0.105 | |
| Rash | 43 | 22.5% | 17 | 18.5% | 0.437 | | 1 | 0.5% | 0 | 0.0% | >0.999 | |

In the elderly group, patients undergoing monotherapy experienced less all-grade constipation, nausea/vomiting, peripheral neuropathy, and fatigue than those undergoing combination therapy (Table 4). There were no significant differences in severe adverse events between groups (50.0% vs. 63.2%, *p* = 0.211).

### 3.4. Factors Affecting Survival

The elderly group had a slightly shorter median OS than the non-elderly group (12.2 (95% confidence interval (CI): 9.7–14.5) months vs. 13.0 (95% CI: 10.8–15.1) months; *p* = 0.036) (Figure 1a). Median PFS was also shorter in the elderly group (5.8 (95% CI: 4.0–7.6) months vs. 7.3 (95% CI: 6.0–8.7) months; *p* = 0.005) (Figure 1b).

Within the elderly group, median OS in the monotherapy group was shorter than the combination therapy group (10.4 (95% CI: 6.2–14.6) months vs. 14.1 (95% CI: 11.5–16.8) months; *p* = 0.010) (Figure 2a). The difference in median PFS was not significant (4.5 (95% CI: 2.3–6.7) months vs. 6.7 (95% CI: 4.6–8.9) months; *p* = 0.161) (Figure 2b).

An age of 75 years or older was a significant predictor of OS in the overall cohort in the univariate analysis (hazard ratio (HR): 1.33; *p* = 0.039) but did not remain significant in the multivariate analysis (Table 5). Multivariate Cox regression analyses revealed that NLR values less than 3, mGPS of 0, normal carcinoembryonic antigen (CEA), and choice of triplet therapy with gemcitabine, cisplatin, and S-1 were significant predictors of longer OS.

In the elderly cohort, choice of monotherapy was significantly associated with shorter OS (HR: 1.78, *p* = 0.012), but did not remain significant in multivariate analysis (Table 6). Only mGPS values of 1 or 2 and a CEA of 5 of more were significant predictors of shorter OS.

An age of 75 years or older was a significant predictor of shorter PFS in the overall cohort, in both univariate (HR: 1.47; *p* = 0.006) and multivariate analyses (HR: 1.44; *p* = 0.029) (Table 7). Other significant independent predictors of shorter PFSs were existence of liver metastases, existence of lung metastases, and NLR of 3 or more.

**Table 4.** Adverse events—elderly cohort.

| | All Grades | | | | | Grades 3–4 | | | | |
|---|---|---|---|---|---|---|---|---|---|---|
| **(*n*, %)** | **Mono-Therapy** | | **Combination** | | *p*-Value | **Mono-Therapy** | | **Combination** | | *p*-Value |
| | **(*n* = 54)** | | **(*n* = 38)** | | | **(*n* = 54)** | | **(*n* = 38)** | | |
| All adverse events | 54 | 100.0% | 38 | 100.0% | - | 27 | 50.0% | 24 | 63.2% | 0.211 |
| Hematologic adverse events | | | | | | | | | | |
| Leukopenia | 31 | 57.4% | 29 | 76.3% | 0.061 | 6 | 11.1% | 6 | 15.8% | 0.543 |
| Neutropenia | 36 | 66.7% | 31 | 81.6% | 0.113 | 16 | 29.6% | 17 | 44.7% | 0.137 |
| Anemia | 52 | 96.3% | 38 | 100.0% | 0.510 | 9 | 16.7% | 10 | 26.3% | 0.260 |
| Thrombocytopenia | 40 | 74.1% | 30 | 78.9% | 0.589 | 2 | 3.7% | 4 | 10.5% | 0.226 |
| Febrile neutropenia | 0 | 0.0% | 0 | 0.0% | - | 0 | 0.0% | 0 | 0.0% | - |
| Non-hematologic adverse events | | | | | | | | | | |
| Stomatitis | 5 | 9.3% | 4 | 10.5% | >0.999 | 0 | 0.0% | 0 | 0.0% | - |
| Decreased appetite | 12 | 22.2% | 9 | 23.7% | 0.869 | 0 | 0.0% | 1 | 2.6% | 0.413 |
| Diarrhea | 6 | 11.1% | 10 | 26.3% | 0.058 | 0 | 0.0% | 0 | 0.0% | - |
| Constipation | 23 | 42.6% | 31 | 81.6% | <0.001 | 0 | 0.0% | 0 | 0.0% | - |
| Nausea/vomiting | 12 | 22.2% | 18 | 47.4% | 0.011 | 0 | 0.0% | 0 | 0.0% | - |
| Peripheral neuropathy | 4 | 7.4% | 10 | 26.3% | 0.013 | 0 | 0.0% | 0 | 0.0% | - |
| Alopecia | 1 | 1.9% | 4 | 10.5% | 0.156 | 0 | 0.0% | 0 | 0.0% | - |
| Fatigue | 34 | 63.0% | 36 | 94.7% | <0.001 | 0 | 0.0% | 2 | 5.3% | 0.168 |
| Elevated transaminases | 47 | 87.0% | 35 | 92.1% | 0.515 | 4 | 7.4% | 1 | 2.6% | 0.400 |
| Decreased renal function | 19 | 35.2% | 9 | 23.7% | 0.238 | 1 | 1.9% | 0 | 0.0% | >0.999 |
| Interstitial pneumonitis | 1 | 1.9% | 1 | 2.6% | >0.999 | 1 | 1.9% | 1 | 2.6% | >0.999 |
| Rash | 9 | 16.7% | 8 | 21.1% | 0.594 | 0 | 0.0% | 0 | 0.0% | - |

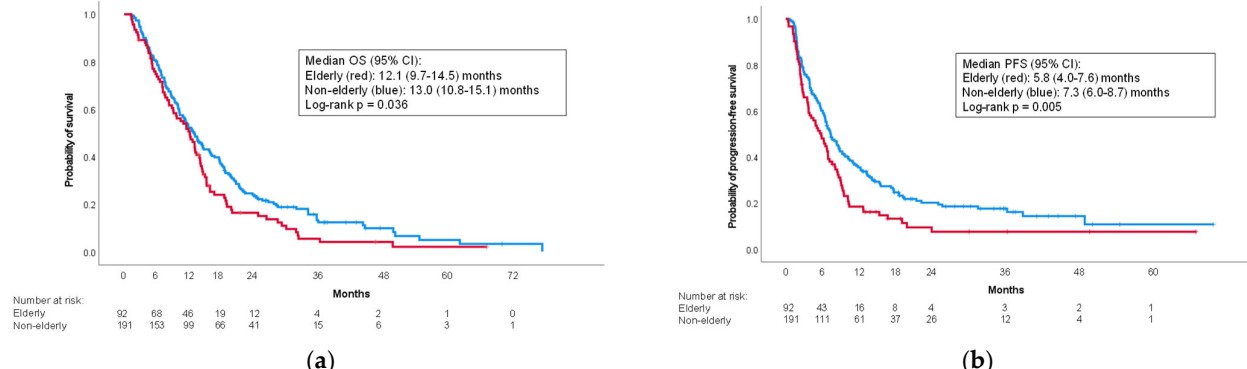

**Figure 1.** Kaplan–Meier curves for the overall cohort. (**a**) Overall survival; (**b**) progression-free survival. CI: confidence interval; OS: overall survival; PFS: progression-free survival.

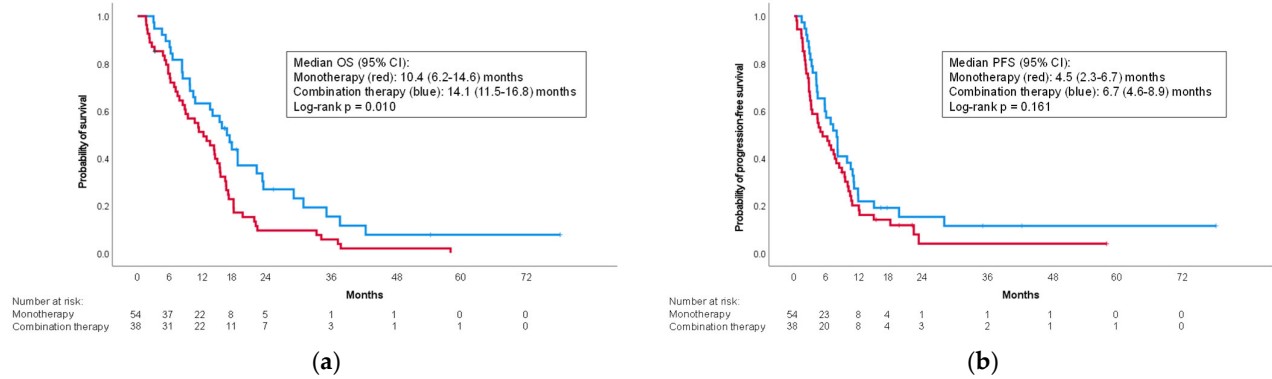

**Figure 2.** Kaplan–Meier curves for the elderly cohort. (**a**) Overall survival; (**b**) progression-free survival. CI: confidence interval; OS: overall survival; PFS: progression-free survival.

**Table 5.** Factors affecting overall survival—overall cohort.

| | Univariate | | | Multivariate (Predictors Only) | | |
|---|---|---|---|---|---|---|
| | Hazard Ratio | 95% CI | *p*-Value | Hazard Ratio | 95% CI | *p*-Value |
| Baseline characteristics | | | | | | |
| Male sex | 0.98 | 0.76–1.27 | 0.904 | | | |
| Elderly (75 years or older) | 1.33 | 1.02–1.73 | 0.039 | 1.07 | 0.78–1.48 | 0.669 |
| Performance status (1 or 2) | 1.51 | 1.12–2.02 | 0.008 | 1.03 | 0.74–1.44 | 0.859 |
| Tumor characteristics | | | | | | |
| Locally advanced (vs. metastatic or recurrence) | 0.58 | 0.40–0.84 | 0.003 | 0.93 | 0.59–1.49 | 0.769 |
| Gallbladder cancer | 1.57 | 1.18–2.09 | 0.002 | 1.30 | 0.94–1.79 | 0.114 |
| Extrahepatic (perihilar) cholangiocarcinoma | 0.80 | 0.60–1.06 | 0.120 | 1.04 | 0.75–1.45 | 0.808 |
| Liver metastasis | 1.33 | 1.03–1.73 | 0.031 | 1.40 | 1.02–1.92 | 0.038 |
| Lung metastasis | 1.34 | 0.94–1.91 | 0.108 | | | |
| Lymph node metastasis | 1.37 | 1.05–1.79 | 0.021 | 1.28 | 0.94–1.74 | 0.113 |
| Peritoneal dissemination metastasis | 1.48 | 1.12–1.96 | 0.006 | 1.36 | 0.99–1.86 | 0.057 |
| Bone metastasis | 2.11 | 0.99–4.50 | 0.052 | | | |
| Laboratory values | | | | | | |
| Neutrophil-to-lymphocyte ratio (3 or more) | 1.69 | 1.31–2.17 | <0.001 | 1.57 | 1.20–2.04 | <0.001 |
| mGPS (1 or 2) | 1.87 | 1.45–2.42 | <0.001 | 1.65 | 1.25–2.17 | <0.001 |
| CEA (5 ng/mL or more) | 2.06 | 1.59–2.65 | <0.001 | 1.73 | 1.32–2.27 | <0.001 |
| CA19-9 (500 U/mL or more) | 1.48 | 1.15–1.92 | 0.003 | 1.18 | 0.90–1.55 | 0.243 |
| Treatment | | | | | | |
| Monotherapy | 1.73 | 1.31–2.28 | <0.001 | 1.39 | 0.97–1.99 | 0.074 |
| First-line GCS | 0.39 | 0.20–0.76 | 0.005 | 0.43 | 0.22–0.85 | 0.016 |
| Any second-line chemotherapy | 0.69 | 0.54–0.89. | 0.005 | | | |
| Conversion surgery | 0.21 | 0.11–0.40 | <0.001 | | | |

CA19-9: carbohydrate antigen; CEA: carcinoembryonic antigen; CI: confidence interval; GCS: gemcitabine + cisplatin + S-1 triplet chemotherapy; mGPS: modified Glasgow prognostic score.

**Table 6.** Factors affecting overall survival—elderly cohort.

| | Univariate | | | Multivariate (Predictors only) | | |
|---|---|---|---|---|---|---|
| | Hazard Ratio | 95% CI | *p*-Value | Hazard Ratio | 95% CI | *p*-Value |
| Baseline characteristics | | | | | | |
| Male sex | 1.01 | 0.65–1.57 | 0.980 | | | |
| Performance status (1 or 2) | 1.13 | 0.72–1.78 | 0.587 | | | |
| Tumor characteristics | | | | | | |
| Recurrence | 1.21 | 0.78–1.88 | 0.386 | | | |
| Extrahepatic (perihilar) cholangiocarcinoma | 0.57 | 0.35–0.94 | 0.021 | 1.01 | 0.62–1.66 | 0.971 |
| Liver metastasis | 1.80 | 1.14–2.84 | 0.140 | | | |
| Lung metastasis | 1.42 | 0.76–2.63 | 0.268 | | | |
| Lymph node metastasis | 1.19 | 0.74–1.92 | 0.470 | | | |
| Peritoneal dissemination metastasis | 1.12 | 0.68–1.82 | 0.667 | | | |
| Bone metastasis | 3.41 | 0.46–25.2 | 0.230 | | | |
| Laboratory values | | | | | | |
| Neutrophil-to-lymphocyte ratio (3 or more) | 1.59 | 1.03–2.45 | 0.036 | 1.28 | 0.80–2.04 | 0.303 |
| mGPS (1 or 2) | 2.35 | 1.51–3.67 | <0.001 | 2.24 | 1.42–3.54 | <0.001 |
| CEA (5 ng/mL or more) | 1.87 | 1.20–2.92 | 0.006 | 1.70 | 1.03–2.79 | 0.036 |
| CA19-9 (37 U/mL or more) | 0.90 | 0.56–1.43 | 0.651 | | | |
| Treatment | | | | | | |
| Monotherapy | 1.78 | 1.14–2.78 | 0.012 | 1.48 | 0.93–2.35 | 0.102 |
| Any second-line chemotherapy | 0.62 | 0.40–0.97 | 0.036 | | | |
| Conversion surgery | 0.12 | 0.17–0.90 | 0.040 | | | |

CA19-9: carbohydrate antigen; CEA: carcinoembryonic antigen; CI: confidence interval; GCS: gemcitabine + cisplatin + S-1 triplet chemotherapy; mGPS: modified Glasgow prognostic score.

**Table 7.** Factors affecting progression-free survival—overall cohort.

| | Univariate | | | Multivariate | | |
|---|---|---|---|---|---|---|
| | Hazard Ratio | 95% CI | *p*-Value | Hazard Ratio | 95% CI | *p*-Value |
| Baseline characteristics | | | | | | |
| Male sex | 1.01 | 0.78–1.32 | 0.916 | | | |
| Elderly (75 years or older) | 1.47 | 1.12–1.94 | 0.006 | 1.44 | 1.04–1.99 | 0.029 |
| Performance status (1 or 2) | 1.21 | 0.89–1.65 | 0.233 | | | |
| Tumor characteristics | | | | | | |
| Locally advanced (vs. metastatic or recurrence) | 0.47 | 0.31–0.70 | <0.001 | 0.69 | 0.42–1.18 | 0.130 |
| Gallbladder cancer | 1.45 | 1.07–1.92 | 0.017 | 1.21 | 0.87–1.68 | 0.252 |
| Liver metastasis | 1.49 | 1.13–1.96 | 0.005 | 1.47 | 1.06–2.04 | 0.023 |
| Lung metastasis | 1.60 | 1.11–2.29 | 0.016 | 1.69 | 1.15–2.47 | 0.007 |
| Lymph node metastasis | 1.33 | 1.01–1.76 | 0.041 | 1.25 | 0.91–1.70 | 0.167 |
| Peritoneal dissemination metastasis | 1.36 | 1.03–1.82 | 0.034 | 1.28 | 0.93–1.77 | 0.133 |
| Bone metastasis | 1.51 | 0.67–3.39 | 0.325 | | | |
| Laboratory values | | | | | | |
| Neutrophil-to-lymphocyte ratio (3 or more) | 1.69 | 1.30–2.20 | <0.001 | 1.64 | 1.24–2.16 | <0.001 |
| mGPS (1 or 2) | 1.63 | 1.25–2.14 | <0.001 | 1.31 | 0.98–1.75 | 0.072 |
| CEA (5 ng/mL or more) | 1.59 | 1.22–2.07 | <0.001 | 1.27 | 0.96–1.69 | 0.101 |
| CA19-9 (500 ng/mL or more) | 1.55 | 1.18–2.02 | <0.001 | 1.29 | 0.97–1.72 | 0.084 |
| Treatment | | | | | | |
| Monotherapy | 1.59 | 1.19–2.13 | 0.002 | 1.20 | 0.85–1.69 | 0.313 |
| First-line GCS | 0.52 | 0.29–0.93 | 0.028 | 0.56 | 0.30–1.02 | 0.060 |

CA19-9: carbohydrate antigen; CEA: carcinoembryonic antigen; CI: confidence interval; GCS: gemcitabine + cisplatin + S-1 triplet chemotherapy; mGPS: modified Glasgow prognostic score.

In the elderly cohort, monotherapy was not a significant predictor of PFS (Table 8). Significant independent predictors of shorter PFS were existence of liver metastases, existence of lung metastases, mGPS values of 1 or 2, and elevated CEA.

**Table 8.** Factors affecting progression-free survival—elderly cohort.

| | Univariate | | | Multivariate (Predictors Only) | | |
|---|---|---|---|---|---|---|
| | Hazard Ratio | 95% CI | *p*-Value | Hazard Ratio | 95% CI | *p*-Value |
| Baseline characteristics | | | | | | |
| Male sex | 1.17 | 0.75–1.83 | 0.494 | | | |
| Performance status (1 or 2) | 0.97 | 0.61–1.57 | 0.914 | | | |
| Tumor characteristics | | | | | | |
| Locally advanced (vs. metastatic or recurrence) | 0.40 | 0.20–0.80 | 0.010 | 0.64 | 0.28–1.43 | 0.274 |
| Extrahepatic (perihilar) cholangiocarcinoma | 0.46 | 0.43–0392 | 0.022 | 0.67 | 0.39–1.15 | 0.150 |
| Liver metastasis | 2.61 | 1.63–4.19 | <0.001 | 2.10 | 1.26–3.47 | 0.003 |
| Lung metastasis | 2.40 | 1.27–4.52 | 0.007 | 3.10 | 1.54–6.19 | 0.001 |
| Lymph node metastasis | 1.01 | 0.62–1.65 | 0.977 | | | |
| Peritoneal dissemination metastasis | 1.14 | 0.70–1.87 | 0.595 | | | |
| Bone metastasis | 2.81 | 0.38–20.7 | 0.310 | | | |
| Laboratory values | | | | | | |
| Neutrophil-to-lymphocyte ratio (3 or more) | 1.58 | 1.02–2.46 | 0.042 | 1.10 | 0.66–1.85 | 0.715 |
| mGPS (1 or 2) | 1.69 | 1.07–2.66 | 0.023 | 2.07 | 1.24–3.45 | 0.006 |
| CEA (5 ng/mL or more) | 2.52 | 1.59–4.01 | <0.001 | 1.87 | 1.05–3.35 | 0.035 |
| CA19-9 (37 ng/mL or more) | 0.97 | 0.60–1.57 | 0.901 | | | |
| Treatment | | | | | | |
| Monotherapy | 1.38 | 0.88–2.16 | 0.163 | | | |

CA19-9: carbohydrate antigen; CEA: carcinoembryonic antigen; CI: confidence interval; GCS: gemcitabine + cisplatin + S-1 triplet chemotherapy; mGPS: modified Glasgow prognostic score.

## 4. Discussion

In this study, we conducted a retrospective review of chemotherapy for BTC patients in the pre-immune checkpoint inhibitor (ICI) era, with a focus on safety and efficacy of palliative chemotherapy in elderly patients. We found that elderly patients had worse ECOG PS and were more likely to receive monotherapy than the non-elderly group; however, the response rates were similar. Both OS and PFS were longer in the non-elderly group; however, severe adverse events were also more frequent. Age was an independent predictor of PFS but not of OS. Within the elderly cohort, patients receiving monotherapy were less likely to proceed to second-line treatment than those receiving combination therapy. There were no differences in severe adverse events between groups. Median OS, but not PFS, was longer in the combination therapy group; however, choice of monotherapy was not an independent predictor of either OS or PFS in the multivariate analyses.

Despite the lack of an upper age limit in most recent prospective studies, elderly patients are grossly underrepresented (Table 9). While a poor PS is more common in the elderly, it is difficult to deny that trial investigators are reluctant to enter even healthy octogenarians into clinical trials. Patients aged 65 years or older, 70 years or older, and 75 years or older made up 74%, 54%, and 33% of the patients in our real-world study, respectively, while 32–64% were aged 65 years or older and 0–17% were aged 75 years or older in prospective studies. We conducted chemotherapy in BTC patients as old as 89 years, while the maximum age from the ten major evaluated studies was 84 years.

**Table 9.** Elderly patient participation in major recent prospective studies.

| Trial Name | Year | Phase | Treatment | Line | n | Upper Age Limit (Inclusion Criteria) | Oldest (Years) | Median Age | ≥65 | ≥70 | ≥75 | PS |
|---|---|---|---|---|---|---|---|---|---|---|---|---|
| FUGA-BT [7] | 2019 | III | GC vs. GS | 1 | 354 | 79 | 79 | 67/67 | 64% | | 17% | 0–1 |
| PRODIGE 12-ACCORD 18-UNICANCER GI [9] | 2019 | III | Gemcitabine/Oxaliplatin vs. observation | Adjuvant | 196 | None | 83 | 63/63 | | | | 0–2 |
| BILCAP [10] | 2019 | III | Capecitabine vs. observation | Adjuvant | 447 | None | 69 | 62/64 | | 0% | 0% | 0–1 |
| ClarIDHy [11] | 2020 | III | Ivodenib vs. placebo | 2 or 3 | 185 | None | 83 | 61/63 | | | | 0–1 |
| FIGHT-202 [12] | 2020 | III | Pemigatinib | 2 | 146 | None | 78 | 59 | 32% | | 8% | 0–2 |
| ABC-06 [13] | 2021 | III | FOLFOX vs. ASC 5-FU/LV ± | 2 | 162 | None | 84 | 65/65 | 50% | | | 0–2 |
| NIFTY [14] | 2021 | IIb | Nanoliposomal irinotecan | 2 | 174 | None | 84 | 63/65 | 48% | | | 0–1 |
| TOPAZ-1 [15] | 2022 | III | GC ± Durvalumab | 1 | 685 | None | 85 | 64/64 | 47% | | | 0–1 |
| KHBO1401-MITSUBA [16] | 2023 | III | GC ± S-1 | 1 | 246 | None | 84 | 68/68 | | | | 0–2 |
| KEYNOTE-966 [17] | 2023 | III | GC ± Pembrolizumab | 1 | 1069 | None | 71 | 64/63 | 47% | | | 0–1 |
| (This study) | 2023 | - | GCS, GC, GS, Gemcitabine, S-1 | 1 | 283 | None | 89 | 70 | 74% | 54% | 33% | 0–2 |

ASC: active symptom control; GC: gemcitabine + cisplatin; GCS: gemcitabine + cisplatin + S-1 triplet chemotherapy; GS: gemcitabine + S-1, PS: performance status.

Gemcitabine monotherapy has been reported to be similarly safe and effective in elderly BTC patients, with cutoffs set at 70 [26] and 75 [20] years of age. With respect to combination therapy, patients aged 70 years or older predicted poor prognoses in a study on BTC patients receiving gemcitabine and S-1 (GS) [27]. On the other hand, age was not a significant predictor of survival in BTC patients receiving gemcitabine and cisplatin (GC) [28]. An analysis of patients receiving either GC or GS in a clinical trial revealed no significant differences in survival or adverse events based on age with a cutoff of 75 years old, although the elderly group only included patients aged 75–79 years old [19].

A long review of studies comparing elderly and non-elderly BTC patients undergoing chemotherapy, including a subgroup analysis of the ABC-02 trial [8], found that age had no

impact on either OS or PFS, regardless of whether monotherapy or combination therapy was provided [29]. The same study found that combination therapy achieved higher OS (HR: 0.54, $p$ = 0.001) and PFS (HR: 0.60, $p$ = 0.004) results in a subgroup of patients aged 70 years old or older.

We found that, while OS and PFS were longer in non-elderly patients, they also experienced more severe adverse events. As severe adverse events can lead to rapid decline in ECOG PS and often lead to the termination of chemotherapy in the elderly, physician judgment in choosing between monotherapy and combination therapy is crucial to maximize OS while maintaining quality of life. Age was not an independent predictor of OS, consistent with previous prospective studies [30], even when physician discretion was introduced. Our study indicated that metrics, such as NLR, mGPS, and CEA, were better predictors of OS than age, implying that judicious selection of chemotherapy regimens can contribute to the achievement of OS in elderly patients comparable to that of non-elderly patients.

Our investigation of differences in outcomes between monotherapy and combination therapy in elderly BTC patients also shed light on the optimal treatment strategy in this population. Specifically, choice of monotherapy was not an independent predictor of neither OS nor PFS in multivariate analysis. Starting with monotherapy also allowed almost 40% of patients to proceed to second-line chemotherapy, which is similar to the percentage of non-elderly patients who received second-line chemotherapy. On the other hand, three elderly patients received combination therapy and went on to receive conversion surgery, achieving prolonged OS. Thus, combination therapy may not necessarily be preferable to monotherapy in patients aged 75 years and over; however, some fit patients, such as those who were candidates for conversion therapy, may have benefitted from aggressive combination therapy. Elderly patients with good PS and favorable baseline characteristics predicting longer PFS in this study (absence of liver or lung metastases, NLR < 3, and CEA < 5) may also benefit from combination therapy, regardless of age. A trial comparing combination therapy at a reduced dose to full-dose monotherapy is ongoing for elderly patients with pancreatic cancer [31], and similar studies may be beneficial for elderly BTC patients.

Systemic therapy for BTC is undergoing a paradigm shift towards ICIs and targeted therapy at present, with many clinical trials underway [32]. Data on elderly patients gained in this study may serve as a comparison arm for future real-world analyses involving such new agents. While treatment for BTC has finally entered the ICI era with the TOPAZ-1 trial [15], various questions remain unanswered. For example, it remains unclear whether elderly patients should be given GC at a reduced dose to allow for combination therapy with ICIs [15,17]. Another alternative to be considered is whether or not gemcitabine monotherapy can be combined with ICIs in elderly patients. As most prospective studies have neglected to perform subgroup analyses based on age [26], more research is needed to serve unmet needs for safe and effective treatment in elderly BTC patients.

This study had several limitations. This was a single-center, retrospective study with inherent selection bias. BTC is a heterogeneous disease with varying characteristics, limiting the applicability of our results to underrepresented cancer types, such as ampullary cancer. Analyses of co-morbidities and geriatric assessment were not conducted. Data on relative dose intensities were not available. The inclusion of patients receiving S-1 may limit the generalizability of our results to non-Asian countries where the drug is not the standard of care or is not available.

## 5. Conclusions

Despite the use of different chemotherapy regimens, age was not an independent predictor of OS in patients undergoing chemotherapy for BTC. Use of monotherapy vs. combination therapy also did not independently predict OS in BTC patients aged 75 years old or older. While monotherapy appears to be a viable alternative in elderly BTC patients, treatment should be tailored to the individual. Ongoing and future studies involving ICIs and targeted agents may provide safer and more tolerable options for this population.

**Author Contributions:** Conceptualization, T.O. and N.S.; methodology, T.O.; software, T.O.; valida-tion, T.T., T.H. (Tsuyoshi Hamada), T.M. and N.S.; formal analysis, T.O.; investigation, T.O., T.T., T.M., T.I., M.Y. and H.N.; resources, T.O., T.T., T.H. (Tsuyoshi Hamada) and N.S.; data curation, T.O., T.T., T.S., T.H. (Tsuyoshi Hamada), T.M., T.I., M.Y., H.N., T.H. (Tatsuki Hirai), T.F., A.K., M.O. and N.S.; writing—original draft preparation, T.O.; writing—review and editing, T.O., T.T., T.S., T.H. (Tsuyoshi Hamada), T.M., T.I., M.Y., H.N., T.H. (Tatsuki Hirai), T.F., A.K., M.O. and N.S.; visualization, T.O.; supervision, N.S.; project administration, T.T., T.H. (Tsuyoshi Hamada) and N.S. All authors have read and agreed to the published version of the manuscript.

**Funding:** This research received no external funding.

**Institutional Review Board Statement:** The study was conducted in accordance with the Declaration of Helsinki and approved by the Institutional Review Board of Cancer Institute Hospital of Japanese Foundation for Cancer Research (protocol code 2023-GB-016, approved on 16 June 2023).

**Informed Consent Statement:** Patient consent was waived due to the retrospective nature of the study.

**Data Availability Statement:** Data are available from the corresponding author upon reasonable request.

**Conflicts of Interest:** N.S. received scholarship donations from Taiho Pharmaceutical and Chugai Pharmaceutical towards his institution and honoraria from Taiho Pharmaceutical, Eisai, Chugai Pharmaceutical, Takeda Pharmaceutical, and AstraZeneca. All other authors declare no conflict of interest.

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
