# Peer review of "Safety and Effectiveness of Chemotherapy in Elderly Biliary Tract Cancer Patients"

_curroncol, doi:10.3390/curroncol30080524_

Round 1

Reviewer 1 Report

Okamoto et. al performed a retrospective study using their own prospectively maintained database to investigate safety and efficacy of chemotherapy in elderly patients with BTC (biliary tract cancer). Their definition of elderly is greater than 75 year old and total sample size was 283 patients with 91 (32.5%) being greater than 75. They concluded that elderly patients received monotherapy more often than combination therapy compared to non-elderly patients. But with multivariate analysis did not demonstrate the selection of monotherapy was not an independent predictor of shorter OS/PFS. 

Overall it is very well written with very interesting topics, as elderly populations are increasing worldwide and we have often encountered this patient population with BTC. However, there are several concerns as below. 

Major: 

1. If authors claim that this study to be real world data, age greater than 75 is not truly elderly in the current era. In this Reviewer's practice, if patients older than 75 with excellent Performance Status and limited co-morbidities, they would receive the standard combination systemic therapy if indicated. Hence, age 75 seems to be rather arbitrary. Age greater than 80 seems to be more appropriate cut off to this Reviewer. If Authors continue to use this cut off, they should provide much stronger argument than the current one. 

2. Although Authors mentioned this in the Limitation, it should be more emphasized that BTC are extremely heterogeneous cancer as the management is often very different as well. It is unclear that Authors should include ampullary cancer to be BTC (It was included in ABC-02 trial, but not in BILCAP). 

3. This Reviewer agreed to the Author's discussion. With introduction of targeted therapy and ICI therapies in BTC, it might be easier for elderly patients to tolerate these therapies in conjunction with conventional chemotherapies either monotherapy or dose reduced combination. This would require further studies. 

Reviewer 2 Report

This is an interesting study on the role of Chemotherapy in elderly patients with biliary tract cancer. Overall  it is well written.

Comments to the authors:

1. line 78, response to chemotherapy:

have you looked also at the response of tumour markers levels, such as CA 19-9 and CEA?

2. as combination therapy provided better overall survival vs. monotherapy in the elderly group, could the authors propose criteria for more extensive use of combination chemotherapy in the discussion section?

minor corrections are required
